# The Relationship Between Systemic Inflammatory Index and Other Inflammatory Markers with Clinical Severity of the Disease in Patients with Parkinson’s Disease

**DOI:** 10.3390/biomedicines13082029

**Published:** 2025-08-20

**Authors:** Aybala Neslihan Alagoz, Aydan Dagdas, Sena Destan Bunul, Guldeniz Cetin Erci

**Affiliations:** Department of Neurology, Faculty of Medicine, Kocaeli University, Kocaeli 4100, Turkey; aybalaalagoz@hotmail.com (A.N.A.); aydandemir92.ad@gmail.com (A.D.); guldenizcetinerci@gmail.com (G.C.E.)

**Keywords:** Parkinson’s disease, neuroinflammation, biomarker, systemic inflammation

## Abstract

**Background/Objectives**: Parkinson’s disease (PD) is a progressive neurodegenerative disorder characterized by the loss of dopaminergic neurons in the substantia nigra (SN), pathological accumulation of alpha-synuclein, and chronic neuroinflammation. The aim of this study is to evaluate the serum levels of systemic inflammatory markers such as neutrophil–lymphocyte ratio (NLR), neutrophil-HDL ratio (NHR), monocyte-HDL ratio (MHR), platelet–lymphocyte ratio (PLR), IL-6, IGF-1, systemic immune-inflammation index (SII), and systemic inflammation response index (SIRI) in patients with PD, and to analyze the relationship between these markers and the clinical stage of the disease as well as its motor and non-motor symptoms. **Methods**: Fifty-one patients diagnosed with PD and forty-nine HC matched for age and sex were evaluated prospectively. **Results**: NLR, NHR, and IGF-1 levels were found to be significantly higher in the PD group compared to the HC group (*p* < 0.05). There was no significant difference between the two groups in terms of PLR, MHR, SII, and SIRI. No significant relationship was found between the inflammatory markers and disease duration, clinical scales, or symptoms. **Conclusions**: These findings support the role of systemic inflammation in the pathophysiology of PD. Further multi-center, long-term follow-up studies—including simultaneous measurements of central nervous system inflammation markers—are needed for translation into clinical practice.

## 1. Introduction

Parkinson’s disease (PD) is a progressive neurodegenerative movement disorder characterized by the loss of dopaminergic neurons in the substantia nigra (SN), accumulation of alpha-synuclein (α-Syn), and chronic neuroinflammation. Clinically, PD presents with motor symptoms, such as resting tremor, bradykinesia, and rigidity, as well as non-motor symptoms, including autonomic dysfunction, incontinence, and cognitive impairment [1,2]. The lifetime incidence increases five- to tenfold between the sixth and ninth decades of life. The prevalence is approximately 2% among individuals over the age of 65 and rises to 4% in those aged 85 and older [3]. Consequently, in modern societies with growing elderly populations, PD represents a major public health challenge that diminishes the quality of life, impairs physical function, and contributes to mortality through various complications.

Neuroinflammation has gained increasing attention in the pathogenesis of PD. Although the inflammatory response triggered by microglial cell activation in the central nervous system (CNS) contributes to neuronal damage [4], the role of systemic inflammation (SI) remains unclear. Recent studies suggest that communication between the immune system and the CNS occurs via the glymphatic system, meningeal lymphatic vessels, or a disrupted blood–brain barrier (BBB) [5,6,7]. Although it is hypothesized that immune cells activated in the periphery may affect the CNS via these pathways, current evidence is insufficient to substantiate this hypothesis.

Given the systemic inflammatory processes underlying chronic diseases, simple, cost-effective, and reproducible inflammatory markers are gaining clinical relevance. Parameters such as the neutrophil-to-lymphocyte ratio (NLR), platelet-to-lymphocyte ratio (PLR), systemic immune-inflammation index (SII), and systemic inflammation response index (SIRI), derived from routine hemogram tests, are considered indirect yet highly sensitive indicators of SI [8,9].

High-density lipoprotein (HDL), known for its antioxidant and anti-inflammatory properties, is inversely associated with SI and endothelial dysfunction [10]. Recent studies have demonstrated that the neutrophil-to-HDL ratio (NHR) and monocyte-to-HDL ratio (MHR) are reliable indicators of systemic inflammatory activity [11]. Similarly, in neurological diseases, parameters such as NLR, NHR, and PLR have been accepted as important inflammatory markers in both clinical staging of the disease and treatment monitoring [12,13,14].

Interleukin-6 (IL-6), a key marker of SI is a proinflammatory cytokine involved in amyloid plaque formation and tau phosphorylation [15]. The neuronal peptide hormone insulin-like growth factor-1 (IGF-1) is considered critical for regulating cerebral blood flow, neurogenesis, and neuroplasticity. Serum IGF-1 levels, which inhibit amyloid aggregation and tau phosphorylation, have been positively associated with the risk of developing PD [16].

This study aimed to evaluate the serum levels of systemic inflammatory markers, such as NLR, NHR, MHR, PLR, IL-6, IGF-1, HDL, SII, and SIRI, in patients with PD and to investigate their association between these markers and the clinical stage of the disease, as well as motor and non-motor symptoms. The data obtained are expected to contribute to a better understanding of the inflammatory mechanisms in PD and to the clinical integration of inflammation-based biomarkers.

## 2. Materials and Methods

### 2.1. Study Design and Patient Selection

In this cross-sectional study, blood samples obtained at the initial presentation of patients evaluated at the PD outpatient clinic over a one-year period were analyzed. Hematological parameters of patients diagnosed by two neurologists specialized in movement disorders were compared with those of age- and sex-matched healthy controls (HC). The study was approved by the Ethics Committee of Kocaeli University Faculty of Medicine (Ethics Committee File No: GOKAEK 2022/06.01, Approval Date: 28 March 2022).

The inclusion criteria for the patient group were adults aged ≥ 40 years with a clinical diagnosis of PD based on the UK Parkinson’s Disease Society Brain Bank criteria, and who provided informed consent. The exclusion criteria were as follows: history of rheumatologic, immunologic, or malignant diseases; current immunosuppressive or chemotherapeutic treatment; recent infection or antibiotic use within the past three months; blood transfusion within the past six months; use of antiplatelet agents (e.g., acetylsalicylic acid or clopidogrel); chronic liver or kidney disease; and use of nonsteroidal anti-inflammatory drugs within the last two weeks. For the HC group, the inclusion criteria were healthy adults aged ≥ 40 years without any known neurological, inflammatory, autoimmune, or malignant diseases, and who also provided informed consent. The same exclusion criteria were applied to both groups.

### 2.2. Demographic and Clinical Characteristics of Patients

Demographic data (age and sex), medical and family history, body mass index (BMI), time since diagnosis, and levodopa treatment status were recorded for all patients. The clinical characteristics of the patients, their medical treatments, the presence of motor and non-motor symptoms, and the Unified Parkinson’s Disease Rating Scale (UPDRS) were evaluated in this study. Disease severity was determined using the Modified Hoehn–Yahr Scale (MHYS) staging system. The standardized Mini-Mental Test (sMMT) was used to assess cognitive function in the patient and HC groups participating in the study.

### 2.3. Serum Inflammatory Markers

Two milliliters of venous blood were collected from all participants between 8:00 and 9:00 a.m. after a 12 h fast and transferred into tubes containing ethylenediaminetetraacetic acid (EDTA). Samples were analyzed within one hour of collection in the hematology laboratory using a Beckman Coulter UniCel DxH 800 analyzer (Beckman Coulter, Wycombe, UK). This analyzer was used to determine white blood cells (WBC), neutrophil, lymphocyte, monocyte, platelet, HDL, IGF-1, and IL-6 levels.

In this study, various systemic inflammatory parameters indicative of SI were calculated. NLR was calculated as the ratio of neutrophils to lymphocytes; MHR as monocytes to HDL; NHR as neutrophils to HDL; and PLR as platelets to lymphocytes. SII was derived by multiplying the platelet count by the neutrophil-to-lymphocyte ratio, and SIRI by multiplying the monocyte count by the neutrophil-to-lymphocyte ratio. Additionally, SII was determined by multiplying the neutrophil-to-lymphocyte ratio by the platelet count (platelet × neutrophil-to-lymphocyte ratio), and the SIRI was determined by multiplying the neutrophil-to-lymphocyte ratio by the monocyte count (monocyte × neutrophil-to-lymphocyte ratio).

### 2.4. Statistical Analysis

Data were analyzed using IBM SPSS version 26. Continuous variables were expressed as mean ± standard deviation, while categorical variables were presented as frequency (*n*) and percentage (%). The statistical power of this study was calculated using the G*Power 3.1.9.7 software. A significance level of α = 0.05 and a medium effect size (Cohen’s d = 0.5) were assumed for the analysis, and the resulting statistical power (1 − β) was calculated to be 80%. Relationships between categorical variables were evaluated using the Chi-square test. The Shapiro–Wilk and Kolmogorov–Smirnov tests were applied to assess the normality of continuous variables. Independent samples *t*-tests were used to compare continuous variables between groups. Repeated measures were analyzed using the General Linear Model (GLM) repeated measures test. Correlations between inflammatory markers and clinical parameters—including disease duration, MHYS, and UPDRS—were assessed using Spearman’s correlation analysis.

## 3. Results

A total of 51 patients diagnosed with PD and 49 age- and sex-matched HC were prospectively enrolled over a 12-month period at the neurology outpatient clinic, in accordance with the study criteria. As detailed in Table 1, the demographic analysis revealed that the mean age of the 100 participants was 66.18 ± 9.2 years. The sex distribution comprised 46% female (*n* = 46) and 54% male (*n* = 54). The average disease duration was 7.53 ± 7.234 years. There were no statistically significant differences in age, sex, or BMI between the PD and HC groups (*p* > 0.05). The mean total UPDRS score was 79.8 ± 40.47, with motor and non-motor scores averaging 41.04 ± 23.48 and 15.61 ± 8.71, respectively. The mean MHYS score was 2.59 ± 1.22. There was no significant difference between groups in the sMMT scores (*p* > 0.05) (Table 1).

Analysis of mean values for inflammatory markers—including neutrophils, lymphocytes, leukocytes, HDL, MHR, PLR, SII, and SIRI—showed no statistically significant differences between the patient and control groups (*p* > 0.05). In contrast, levels of IGF-1, NLR, and NHR were significantly higher in the PD group than in the HC group (*p* < 0.05) (Table 2).

A correlation analysis was conducted to evaluate the relationship between PD non-motor symptoms and inflammatory markers. The levels of IL-6 were found to be significantly associated with nonmotor symptoms, including psychosis, depression, anxiety, apathy, dopamine dysregulation syndrome, constipation, and lightheadedness on standing (LHON). Additionally, significant associations were identified between LHON and the levels of IGF-1, NLR, LMR, and SIRI. PLR was exclusively associated with fatigue. The LMR demonstrated a significant negative correlation with constipation, LHON, and nM-EDLTS (*p* < 0.05) (Table 3).

Significant differences were also observed between PD motor symptoms and several inflammatory markers. Both NLR and PLR correlated significantly with the time spent with dyskinesias (TSWD). Moreover, SIRI was significantly associated with TSWD and painful off-state dystonias (POSD). However, SII showed a significant correlation only with eating tasks (*p* < 0.05). (Table 4)

## 4. Discussion

This study investigated systemic inflammatory markers in patients with PD, highlighting elevated levels of NLR, NHR, and IGF-1 compared to HC. However, our results also revealed a lack of correlation between these markers and disease severity or clinical scores. These findings reinforce the potential role of SI in the pathophysiology of PD.

NLR is a commonly used marker of SI. Elevated NLR values have been associated with cardiometabolic events, cerebrovascular disorders, epilepsy and neurodegenerative diseases such as Alzheimer’s and PD [17,18,19,20,21]. While Akil et al. reported significantly higher NLR levels in PD patients compared to HC [22], other studies have shown inconsistent results [23]. More recent research indicates that both NLR and MLR are positively associated with disease severity and inversely related to disease duration in PD [24,25,26]. PLR has also emerged as a potential indicator of inflammation severity, with some studies reporting elevated levels and significant correlations with MHYS scores [26]. In our study, NLR was significantly elevated in PD patients, whereas MLR and PLR levels did not differ between groups. These findings, considered alongside the existing literature, support the role of NLR, MLR, and PLR as markers of chronic inflammation.

IGF-1 is recognized for its neuroprotective and neuroproliferative effects and is considered a potential biomarker in PD [27,28]. A high density of IGF-1 receptors has been identified in the SN, and these receptors appear to protect neurons from dopamine-induced toxicity [29,30]. However, studies on serum IGF-1 levels in PD have yielded inconsistent results. Some have reported elevated levels in PD patients compared to HC [31,32], while others found no significant differences [30,33]. Additionally, most studies, including ours, have not demonstrated any correlation between IGF-1 levels and disease duration or severity [33,34]. These findings suggest that IGF-1 may reflect underlying pathophysiological mechanisms rather than disease progression. Further longitudinal research is needed to clarify its clinical significance.

IL-6 is a key proinflammatory cytokine involved in CNS inflammation. In the substantia nigra, microglia contribute to neuronal damage by releasing cytokines, such as IL-6, interleukin-1 beta (IL-1β), and tumor necrosis factor-alpha (TNF-α) [35]. However, studies on IL-6 levels in PD patients have produced conflicting results. While some report elevated serum IL-6 levels and associations with disease stage [36,37,38], As observed in our study, other reports have found no significant differences between PD patients and HC, nor any correlation with clinical severity [39,40]. This inconsistency may stem from the heterogeneous nature of inflammation in PD and its sensitivity to external influences such as subclinical infections, undiagnosed comorbidities, or medication effects. Moreover, IL-6′s short half-life and circadian fluctuations may compromise measurement reliability, underscoring the importance of repeated sampling in future research.

HDL cholesterol plays an inhibitory role in neutrophil activation, adhesion, and migration, thereby exhibiting anti-inflammatory effects [41]. Previous studies have shown that dyslipidemia associated with the inflammatory response is effective in the onset and progression of PD. Several studies have shown that NHR is significantly elevated in PD patients compared to HC and may decrease with disease progression [21,42]. In Creutzfeldt–Jakob disease, elevated NHR and MHR levels were observed in patients with basal ganglion hyperintensities on MRI [43]. Similarly, a study comparing patients with multiple system atrophy (MSA), PD, and HC found significantly higher MHR values in the MSA group, while no difference was found between PD patients and HC [44]. In our study, NHR was significantly higher in the PD group, whereas MHR showed no significant difference between groups. Furthermore, the lack of association between NHR and MHR with UPDRS scores suggests that these markers provide limited predictive insight into disease progression. Although NHR may be predictive in PD compared to MHR but definitive conclusions require further investigation.

The SII is a widely recognized, cost-effective, and common biomarker for assessing SI [45]. SIRI is a new inflammatory index that integrates neutrophil, lymphocyte, and monocyte counts and has prognostic value in inflammatory and malignant diseases [46]. Although some studies suggest that elevated SII levels may be associated with an increased risk of developing PD and correlate with motor symptom severity [47,48], evidence remains limited and inconsistent. Moreover, while SII and SIRI have been linked to the prognosis of conditions such as hypertension, stroke, cancer, and metabolic disorders [49,50], their role in neurodegenerative diseases like PD is not well established. In our study, neither SII nor SIRI levels differed significantly between PD patients and HC, nor were they associated with clinical stage and symptom severity. These findings indicate that, despite their recognized value in systemic inflammatory conditions, SII and SIRI may have limited applicability as biomarkers for PD.

Several studies have linked peripheral inflammatory markers with both motor [51,52] and non-motor symptoms [53,54] in PD. Although we did not observe a significant correlation between inflammatory markers and the total UPDRS score, consistent with the literature, detailed analyses of its individual subscales revealed significant associations, particularly with non-motor symptoms. These findings suggest that SI may contribute to both motor and non-motor manifestations of PD.

### Limitations

This study is limited by its single-center, cross-sectional design, moderate sample size, and the lack of control for lifestyle factors that may influence inflammatory markers. Undetected subclinical conditions and variability in disease stage may have influenced results, while the temporal variability of inflammatory markers highlights the need for longitudinal assessment.

## 5. Conclusions

In this study, SI markers in patients with PD showed significantly elevated levels of NLR, NHR, and IGF-1 compared to HC. In contrast, no significant differences were observed in PLR, MHR, HDL, IL-6, SII, or SIRI levels. There were no significant associations between inflammatory markers and clinical scales, disease duration, or symptom severity. These findings support the involvement of SI in the pathophysiology of PD. Further large-scale, longitudinal studies—ideally incorporating both peripheral and central inflammatory markers—are needed to clarify the temporal dynamics of SI and its potential role in disease progression or therapeutic monitoring in PD.

## Figures and Tables

**Table 1 biomedicines-13-02029-t001:** Sociodemographic and Clinical Characteristics of the Patients.

Parameters	Control	Patient	*p*
Age (years)	63.59	68.67	>0.05
Sex (f: female, m: male)	f = 24 m = 25	f = 22 m = 29	>0.05
BMI (kg/m^2^)	26.19	27.80	>0.05
Disease duration (year, mean ± SD)	-	7.53 ± 7.234	-
Levodopa treatment (%)	-	64.7	-
sMMT (mean ± SD)	26.27 ± 5.404	26.10 ± 3.270	>0.05
MHYS (mean ± SD)	-	2.59 ± 1.22	-
Total UPDRS (mean ± SD)	-	79.8 ± 40.474	-
Total Motor UPDRS (mean ± SD)	-	41.04 ± 23.48	-
Total Non-motor UPDRS (mean ± SD)	-	15.61± 8.71	-

BMI: Body Mass Index, UPDRS: Unified Parkinson’s Disease Rating Scale, MHYS: Modified Hoehn–Yahr Scale, sMMT: Standardize Mini Mental Test.

**Table 2 biomedicines-13-02029-t002:** Comparison of Inflammatory Markers Between HC and PD Groups.

Inflammatory Markers	Control	Patient	*p*
Neutrophil (×10^9^/L)	4.24 ± 1.62	4.83 ± 1.80	0.081
Lymphocyte (×10^9^/L)	2.15 ± 0.59	1.94 ± 0.60	0.077
Leukocyte (×10^9^/L)	8.19 ± 6.95	7.57 ± 2.02	0.455
IL-6 (pg/mL)	9.36 ± 7.03	5.45 ± 4.23	0.758
IGF-1 (pg/mL)	107.46 ± 49.89	142.54 ± 66.37	0.005
HDL-C (mmol/L)	50.57 ± 12.83	46.66 ± 10.28	0.096
MHR (×10^9^/mmol)	0.16 ± 0.11	0.16 ± 0.05	0.160
NLR (×10^9^/mmol)	2.06 ± 0.81	2.79 ± 1.60	0.022
NHR (×10^9^/mmol)	0.08 ± 0.04	0.10 ± 0.04	0.030
PLR (×10^9^/mmol)	128.33 ± 42.42	128.09 ± 53.02	0.463
SII (×10^9^/L)	541.08 ± 259.08	659.64 ± 453.96	0.466
SIRI (×10^9^/L)	1.22 ± 0.56	1.61 ± 1.17	0.167

IL-6, Interleukin-6; IGF-1, Insulin-Like Growth Factor-1; HDL, High-Density Lipoprotein; MHR, Monocyte-HDL Ratio; NLR, Neutrophil–Lymphocyte Ratio; NHR, Neutrophil-HDL Ratio; PLR, Platelet–Lymphocyte Ratio; SII, Systemic Immune Inflammation Index; SIRI, Systemic Inflammatory Response Index.

**Table 3 biomedicines-13-02029-t003:** Correlation Analysis of Inflammatory Markers with PD Non-motor Symptoms.

Parameters	Correlation Coefficient (rho)	*p* Value	CI (%95)
IL-6 & psychosis	0.396	<0.06	0.38–0.40
IL-6 & depression	0.191	<0.001	0.18–0.20
IL-6 & anxiety	0.272	<0.001	0.26–0.28
IL-6 & apathy	0.126	<0.001	0.11–0.13
IL-6 & nonmotor DDS	0.324	<0.001	0.29–0.31
IL-6 & constipation	0.359	<0.001	0.35–0.36
IL-6 & LHON	0.49	<0.001	0.48–0.49
IGF-1 & LHON	0.193	0.045	0.52–0.16
NLR & LHON	0.291	0.038	0.15–0.52
PLR & fatigue	−0.280	0.047	0.51–0.12
LMR & constipation	−0.277	0.049	0.51–0.13
LMR & LHON	−0.317	0.023	0.54–0.14
LMR & nM-EDLTS	−0.304	0.030	0.50–0.12
SIRI & LHON	0.296	0.035	0.19–0.53

DDS, Dopamine Dysregulation Syndrome; IL-6, interleukin-6; LHON, Light Headedness on Standing; IGF-1, Insulin-Like Growth Factor-1; NLR, Neutrophil–Lymphocyte Ratio; PLR, Platelet–Lymphocyte Ratio; LMR, Lymphocyte-Monocyte Ratio; nM-EDLTS, Non-Motor Aspects of Experiences of Daily Living Total Score; SIRI, Systemic Inflammatory Response Index.

**Table 4 biomedicines-13-02029-t004:** Correlation Analysis of Inflammatory Markers with PD-Motor Symptoms.

Parameters	Correlation Coefficient (rho)	*p* Value	CI (%95)
NLR & TSWD	0.313	0.025	0.13–0.54
PLR & TSWD	0.361	0.009	0.09–0.58
SII & Eating Tasks	−0.291	0.039	0.46–0.14
SIRI & TSWD	0.286	0.042	0.09–0.52
SIRI & POSD	0.322	0.021	0.14–0.55

NLR, Neutrophil–Lymphocyte Ratio; PLR, Platelet–Lymphocyte Ratio; SII, Systemic Immune-Inflammation Index; SIRI, Systemic Inflammation Response Index; TSWD, Time Spent With Dyskinesias; POSD, Painful Off State Dystonia.

## Data Availability

The original contributions presented in this study are included in the article. Further inquiries can be directed to the corresponding author.

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
