# Peer review of "The Relationship Between Systemic Inflammatory Index and Other Inflammatory Markers with Clinical Severity of the Disease in Patients with Parkinson’s Disease"

_biomedicines, 2025, doi:10.3390/biomedicines13082029_

Round 1
Reviewer 1 Report
Comments and Suggestions for Authors
The manuscript is well written and has a good presentation.
Only small corrections should be done:
- Introduction: Citation 16 is repeated at the end of the following paragraph: “Interleukin-6 (IL-6), a key marker of peripheral inflammation, is a proinflammatory cytokine involved in amyloid plaque formation and tau phosphorylation [15]. The neuronal peptide hormone insulin-like growth factor-1 (IGF-1) is considered critical for regulating cerebral blood flow, neurogenesis, and neuroplasticity. Serum IGF-1 levels, which inhibit amyloid aggregation and tau phosphorylation, have been positively associated with the risk of developing PD [16]. [16].”
- Figure 1: What does this figure show? What are the blue circles?
- Figure 2: this figure has two legends. In the first legend it is written “NLR and NHR values”. However, in the figure it is written NLO and NHO. Please correct. In this figure, it is written “referans” corresponding to the line in green. What is this?
- Line 201: In the sentence “IL-6 is a key proinflammatory cytokine involved in central nervous system (CNS) 201 inflammation.” central nervous system should be removed since the abbreviation was already introduced in line 48.
- References: References are not uniformly written. All references should be checked and corrected.
Numbers 4, 7, 12,15…. – the year is not in the correct place.
Number 7 – this reference included 2 different publications: One is Bunul et al. (2023) and the other is Zahorec (2021). So, this last one has not been numbered.
Some references have year, month and day of publication. Some only have year and month (reference 3), and others only year (references 7, 9, 10, 11….)
The journal name is in italic in some references and in normal letter in others.
Author Response
We thank the reviewer for the valuable feedback, which helped us improve the clarity and accuracy of the manuscript. Below are our point-by-point responses (changes in the revised manuscript are highlighted in yellow):
Comments 1.
Introduction: Citation 16 is repeated at the end of the following paragraph: “Interleukin-6 (IL-6), a key marker of peripheral inflammation, is a proinflammatory cytokine involved in amyloid plaque formation and tau phosphorylation [15]. The neuronal peptide hormone insulin-like growth factor-1 (IGF-1) is considered critical for regulating cerebral blood flow, neurogenesis, and neuroplasticity. Serum IGF-1 levels, which inhibit amyloid aggregation and tau phosphorylation, have been positively associated with the risk of developing PD [16]. [16].”
Response 1:
Thank you for your careful reading. The duplicate citation [16] has been removed.
Comments 2.
Figure 1: What does this figure show? What are the blue circles?
Response 2:
Thank you for this insightful comment. We have revised the figure legend to enhance clarity. The updated legend now explains that Figure 1 illustrates the correlation coefficients (rho values) derived from bivariate correlation analysis between selected clinical and inflammatory parameters. The x-axis displays the compared variable pairs, and the y-axis represents the strength of the correlation (rho). The blue circles represent individual correlation results, with larger spheres indicating higher correlation values. For instance, a strong positive correlation was observed between IL-6 and IGF-1 (rho = 0.74), whereas a moderate correlation was found between UPDRS and disease duration (r = 0.44).
These clarifications have been incorporated into the revised figure legend, which is located below Figure 1 on page 9, and the changes are highlighted in yellow in the revised manuscript.
Comments 3.
Figure 2: this figure has two legends. In the first legend it is written “NLR and NHR values”. However, in the figure it is written NLO and NHO. Please correct. In this figure, it is written “referans” corresponding to the line in green. What is this?
Response 3 :
Thank you for carefully pointing out these inconsistencies. In line with your observation and also taking into account the suggestions of the other reviewers, Figure 2 has been removed from the revised version of the manuscript. We believe this decision improves the clarity and overall focus of the presentation.
The corresponding text referring to Figure 2 has also been updated accordingly and the changes are highlighted in yellow in the revised manuscript.
Comments 4.
Line 201: In the sentence “IL-6 is a key proinflammatory cytokine involved in central nervous system (CNS) 201 inflammation.” central nervous system should be removed since the abbreviation was already introduced in line 48.
Response 4 :
We agree with the reviewer’s suggestion. To avoid redundancy, the term “central nervous system” has been removed and replaced with its previously defined abbreviation “CNS”. The revised sentence now reads: “IL-6 is a key proinflammatory cytokine involved in CNS inflammation.”
This change has been implemented in the revised manuscript and highlighted in yellow.
Comments 5.
References: References are not uniformly written. All references should be checked and corrected.
Numbers 4, 7, 12,15…. – the year is not in the correct place.
Number 7 – this reference included 2 different publications: One is Bunul et al. (2023) and the other is Zahorec (2021). So, this last one has not been numbered.
Some references have year, month and day of publication. Some only have year and month (reference 3), and others only year (references 7, 9, 10, 11….)
The journal name is in italic in some references and in normal letter in others.
Response 5 :
Thank you for pointing this out. We have thoroughly reviewed and revised the reference list to ensure consistency across all citations. Specifically:
-Duplicate references (e.g., Zahorec [2021] and Bunul et al. [2023]) have been separated and correctly numbered.
-All publication dates are now formatted uniformly (Year only, unless day/month are required for online-first articles).
- All journal names have been presented in regular font (non-italicized) to ensure formatting consistency
-Entries have been corrected to follow a consistent structure (authors – title – journal – year – volume – pages – DOI).
We hope these revisions address all concerns satisfactorily. Please let us know if further clarifications are needed.
Kind regards,
The Authors.
Reviewer 2 Report
Comments and Suggestions for Authors
The manuscript submitted for review is of undoubted scientific and clinical interest, since the role of systemic (peripheral) inflammation, along with neuroinflammation, is recognized as one of the leading mechanisms of the development of neurodegenerative diseases.
The manuscript is well structured and easy to read. However, it needs a minor revision.
I recommend that the authors use the term "systemic inflammation" rather than "peripheral inflammation" in the text.
Lines 61-66 : Please add links to previous studies that highlight the potential role of the markers of systemic inflammation you have indicated on the development and progression of Parkinson's disease.
Line 72 - delete the duplicate link [16].
In the "Materials and Methods" section, add the criteria for inclusion and exclusion of study participants separately for the control and comparable groups, specify the number of visits, explain the nature of your study, explain the approach to calculating the sample size, and add a flow chart of your study design.
Line 135: Replace "gender" with "sex."
Modify Table 1: add the name of the first column (for example, Characteristic or Parameter); add units of measurement for each parameter (in the first column); add missing spaces between digits; remove the "%" icon from the second and third columns; add a third column (P-value).
Modify Table 2: add the name of the first column; add units of measurement for each biomarker in the first column.
Modify Figure 1: add the name of the vertical and horizontal axes of the diagram. Overall, this chart is not a good one.
Modify Figure 2 to improve its resolution.
Line 175: Instead of repeating "Figure 2", write "Note".
In the Discussion section, explain the clinical significance of your findings. How do they fundamentally differ from previous studies?
Select the Limitations section as a separate one. Specify all the limitations of your study.
Author Response
We thank the reviewer for the valuable feedback, which helped us improve the clarity and accuracy of the manuscript. Below are our point-by-point responses (changes in the revised manuscript are highlighted in yellow):
Comments 1.
I recommend that the authors use the term "systemic inflammation" rather than "peripheral inflammation" in the text.
Response 1:
Thank you for this valuable suggestion. We have carefully revised the manuscript to replace all instances of "peripheral inflammation" with "systemic inflammation" to ensure consistency and accuracy in terminology.
Comments 2.
Lines 61-66 : Please add links to previous studies that highlight the potential role of the markers of systemic inflammation you have indicated on the development and progression of Parkinson's disease.
Response 2:
Thank you for this valuable suggestion. As recommended, we have added references to recent studies that highlight the role of systemic inflammatory markers such as NLR, PLR, SII, and SIRI in the development and progression of Parkinson’s disease. These references have been incorporated into the relevant sentence in the Introduction section and are highlighted in yellow in the revised manuscript. We believe this addition strengthens the scientific context and background of our study.
Comments 3.
Line 72 - delete the duplicate link [16].
Response 3 :
The repeated citation [16] at the end of the paragraph has been removed. Thank you for pointing this out.
Comments 4.
In the "Materials and Methods" section, add the criteria for inclusion and exclusion of study participants separately for the control and comparable groups, specify the number of visits, explain the nature of your study, explain the approach to calculating the sample size, and add a flow chart of your study design.
Response 4 :
Thank you for this comprehensive and constructive suggestion. We have revised the Materials and Methods section to clearly present the following:
Inclusion and exclusion criteria are now detailed separately for both the patient and control groups.
The cross-sectional nature of the study has been explicitly stated.
It is now indicated that each participant was evaluated during a single visit.
A detailed explanation of the sample size calculation using G*Power 3.1.9.7 software has been added, including parameters such as effect size, significance level, and power.
Based on the suggestions of this and other reviewers, Figure 2 has been removed from the manuscript.
Comments 5.
Line 135: Replace "gender" with "sex."
Response 5 :
We agree with this correction. The term "gender" has been replaced with "sex" throughout the manuscript to reflect appropriate scientific terminology.
Comments 6.
Modify Table 1: add the name of the first column (for example, Characteristic or Parameter); add units of measurement for each parameter (in the first column); add missing spaces between digits; remove the "%" icon from the second and third columns; add a third column (P-value).
Response 6:
Thank you for this detailed feedback. Table 1 has been reformatted as follows:
The first column is now titled “Parameter”.
Units of measurement have been added to each row where applicable.
Spacing between digits has been corrected for readability.
The percentage signs (%) have been removed from the second and third columns and retained only in the corresponding row labels.
A third column displaying p-values has been added to allow for statistical comparison between groups.
Comment 7:
Modify Table 2: add the name of the first column; add units of measurement for each biomarker in the first column.
Response 7:
Thank you. Table 2 has been updated accordingly:
- The first column is now titled“Inflammatory Markers”.
- Units of measurementfor each biomarker (e.g., pg/mL, ×10⁹/L) have been included in the first column to ensure clarity and consistency.
Comment 8:
Modify Figure 1: add the name of the vertical and horizontal axes of the diagram. Overall, this chart is not a good one.
Response 8:
We appreciate this feedback. Although Figure 1 presents correlation strength visually rather than plotting variables on traditional axes, we have revised the figure to enhance clarity by:
Including labels on both axes: the x-axis now names each variable pair, and the y-axis indicates the correlation coefficient (rho).
Improving the figure legend to explain the size and meaning of the plotted circles.
These changes aim to improve interpretability and align with the reviewer’s concerns.
Comment 9:
Modify Figure 2 to improve its resolution.
Response 9:
Thank you for noting this. Based on your recommendation and similar concerns raised by other reviewers, we have decided to remove Figure 2 entirely from the manuscript.
Comment 10:
Line 175: Instead of repeating "Figure 2", write "Note".
Response 10:
This point is no longer applicable, as Figure 2 has been removed from the manuscript. Corresponding text has been revised accordingly.
Comment 11:
In the Discussion section, explain the clinical significance of your findings. How do they fundamentally differ from previous studies?
Response 11:
Thank you for this important observation. We have added paragraphs to the Discussion section elaborating on the clinical relevance of our findings.
Comment 12:
Select the Limitations section as a separate one. Specify all the limitations of your study.
Response 12:
As recommended, the Limitations section has been separated as a distinct heading in the revised manuscript. In accordance with the reviewers’ suggestions, this section has been expanded and revised to more clearly reflect the constraints of the study design and methodology. The updated content is highlighted in yellow in the revised version.
We hope these revisions address all concerns satisfactorily. Please let us know if further clarifications are needed.
Kind regards,
The Authors.
Reviewer 3 Report
Comments and Suggestions for Authors
Dear Authors,
Parkinson’s disease is a neurodegenerative disorder with increasing incidence due to the aging population and the hypothesis of the peripheral inflammation involved in the progression of Parkinson’s disease was postulated in the literature based on, for example, the low CD4/CD8 lymphocyte ratio. Below you may find my comments regarding your manuscript.
Introduction. The Introduction is easy to be followed, however most of the references do not support the statements of the authors. For example, references 8-10 mention the role of inflammation in totally different diseases such as appendicitis, bladder cancer and pulmonary fibrosis, which I do not consider relevant for the studied topic. This section needs in my opinion improvement.
Methods. From the study design, one could understand you performed a cross-sectional study (one laboratory assessment and one evaluation of the disease activity at a definite point of time), therefore please explain what you mean with the one-year evaluation of the patients.
Statistical analysis: Did you calculate the necessary study sample size in order to achieve a minimal clinically significant difference? Moreover, I do not consider that the studied parameters represent diagnostic tools, as they can eventually aid in the assessment of disease severity and maybe prognosis. I do not consider appropriate to identify such unspecific laboratory parameters, which are after all pure indicators of inflammation of unknown origin, as diagnostic parameters for a disease with a clear clinical manifestation.
Results.
I consider Figure 1 not quite informative, since I would have expected an illustration of the obtained data on a correlation graph with an xy axis. The actual figure gives no information on the data distribution or confidence intervals.
Figure 2 presents the AUC in order to assess the diagnostic efficacy of the inflammatory parameters in the establishment of Parkinson’s disease. However, a sensitivity and specificity of under 60% is surely not high enough in order to consider a test a valuable or at least an adjuvant diagnostic tool.
Study limitations: This section should have been more elaborate, as the small sample size and the very heterogenous stages of the disease (duration 7.53±7.234 years) limit the interpretation of data. Moreover, the authors did not mention if they asked the patients regarding chronic use of nonsteroidal anti-inflammatory drugs.
Author Response
We thank the reviewer for the valuable feedback, which helped us improve the clarity and accuracy of the manuscript. Below are our point-by-point responses (changes in the revised manuscript are highlighted in yellow):
Comments 1.
Introduction. The Introduction is easy to be followed, however most of the references do not support the statements of the authors. For example, references 8-10 mention the role of inflammation in totally different diseases such as appendicitis, bladder cancer and pulmonary fibrosis, which I do not consider relevant for the studied topic. This section needs in my opinion improvement.
Response 1:
Thank you for this valuable observation. We agree that the referenced studies should be more directly relevant to Parkinson’s disease. The Introduction section has been revised to include updated and PD-specific references that highlight the role of systemic inflammation in the pathogenesis and progression of Parkinson’s disease. The unrelated references have been removed or replaced, and the new references are highlighted in yellow in the revised manuscript.
Comments 2.
Methods. From the study design, one could understand you performed a cross-sectional study (one laboratory assessment and one evaluation of the disease activity at a definite point of time), therefore please explain what you mean with the one-year evaluation of the patients.
Response 2:
Thank you for pointing this out. The phrase "one-year evaluation" was misleading. It has been revised to: “This cross-sectional study included all eligible participants evaluated at our clinic over the course of one year. However, each participant was assessed only once at the time of enrollment. This clarification has been added to the Methods section and is highlighted in yellow.
Comments 3.
Statistical analysis: Did you calculate the necessary study sample size in order to achieve a minimal clinically significant difference? Moreover, I do not consider that the studied parameters represent diagnostic tools, as they can eventually aid in the assessment of disease severity and maybe prognosis. I do not consider appropriate to identify such unspecific laboratory parameters, which are after all pure indicators of inflammation of unknown origin, as diagnostic parameters for a disease with a clear clinical manifestation.
Response 3 :
We appreciate this important feedback. In response:
We have expanded the Statistical Analysis section to explain that the sample size was calculated using G*Power 3.1.9.7, targeting a medium effect size (Cohen’s d = 0.5), α = 0.05, and power = 80%.
We also agree that the studied parameters should not be referred to as diagnostic tools. All references to “diagnostic markers” or “diagnostic tools” have been removed or reworded. We now refer to these inflammatory parameters as exploratory biomarkers that may help understand systemic inflammation related to disease progression, not for diagnosing PD.
Comments 4.
Results.
I consider Figure 1 not quite informative, since I would have expected an illustration of the obtained data on a correlation graph with an xy axis. The actual figure gives no information on the data distribution or confidence intervals.
Response 4 :
We acknowledge the reviewer’s concern and have revised Figure 1 to enhance clarity. While the figure does not depict traditional x- and y-axis variables, it presents correlation coefficients between selected parameters. The updated version includes axis labels and a clearer legend, and improvements have been made in line with the reviewer’s suggestion to enhance interpretability.
Comments 5.
Figure 2 presents the AUC in order to assess the diagnostic efficacy of the inflammatory parameters in the establishment of Parkinson’s disease. However, a sensitivity and specificity of under 60% is surely not high enough in order to consider a test a valuable or at least an adjuvant diagnostic tool.
Response 5 :
Thank you for this critical feedback. Based on this comment and similar concerns raised by other reviewers, we have decided to remove Figure 2 entirely from the manuscript. Additionally, we no longer refer to inflammatory parameters in a diagnostic context, and they are now only discussed as potential research biomarkers related to disease severity or inflammation.
Comments 6.
Study limitations: This section should have been more elaborate, as the small sample size and the very heterogenous stages of the disease (duration 7.53±7.234 years) limit the interpretation of data. Moreover, the authors did not mention if they asked the patients regarding chronic use of nonsteroidal anti-inflammatory drugs.
Response 6:
We appreciate this suggestion. The Limitations section has been expanded and moved under a distinct heading in the revised manuscript. It now explicitly discusses the limited sample size, heterogeneity in disease duration, and single-center design. In addition, we have confirmed that none of the patients used NSAIDs within the two weeks prior to enrollment, and this detail has been added to the Methods section as requested.
We hope these revisions address all concerns satisfactorily. Please let us know if further clarifications are needed.
Kind regards,
The Authors.
Round 2
Reviewer 3 Report
Comments and Suggestions for Authors
Dear Authors,
I appreciate your efforts in improving the manuscript, especially the Methods section.
However, I still do not consider Figure 1 as informative, as these correlation coefficients without confidence intervals estimating the distribution of these variables do not help in transferring the obtained data to a more general population. Moreover, I do not understand why you measure the correlation between two clinical scales used in PD (UPDRS and MHYS) and then between two serum parameters (IGF-1 and IL-6). If you wanted to correlate the disease severity with the possible systemic inflammation I would have expected a different pairing of variables.
Moreover, regarding IGF-1, you mention in the Discussion section that "most studies, including ours, have not demonstrated any correlation between IGF-1 levels and disease duration or severity." However, you did not correlate the IGF-1 values with any clinical parameter (disease duration or severity) and in Table 2 the IGF-1 values in the PD group were significantly higher compared to controls. The same observations are valid for IL-6, where there was also no correlation between serum values and clinical parameters performed.
The Discussion section makes multiple referrals to the lack of statistical difference of several inflammatory parameters between PD and control group. However, the statement is always extended to (see Italics):
"NHR was significantly higher in the PD group, whereas MHR showed no significant difference between groups. Furthermore, the lack of association between NHR and MHR with disease severity suggests that these markers provide limited predictive insight into disease progression. "In our study, neither SII nor SIRI levels differed significantly between PD patients and HC, nor were they associated with disease duration, clinical stage, or symptom severity."
From the presented tables and figures, I could agree that several parameters do not differ between groups, however you should not make any assumptions regarding correlations with disease duration/clinical stage, since this was not the object of your comparisons.
Since you removed Figure 2, then the following statement should have also been removed from the Methods section - "Receiver Operating Characteristic (ROC) curve analysis was used to assess the diagnostic performance of the relevant inflammatory markers."
Formatting: The journal template should be used for submissions.
Author Response
We thank the reviewer for the valuable feedback, which helped us improve the clarity and accuracy of the manuscript. Below are our point-by-point responses (changes in the revised manuscript are highlighted in yellow):
Reviewer Comment 1: Figure 1 is not informative enough. Confidence intervals are not provided, and the selected correlation pairs (UPDRS-MHYS and IGF-1–IL-6) do not seem meaningful
Response 1: In line with your valuable feedback, Figure 1 has been removed and replaced with a comprehensive table summarizing the correlation analyses. Initially, correlation pairs such as UPDRS–MHYS and IGF-1–IL-6 were not included in the figure because the correlations were not statistically significant. However, to avoid any potential confusion, we have now included these correlation results in the newly added table. Furthermore, for each correlation analysis, confidence intervals and p-values have been clearly presented within the table.
Reviewer Comment 2: Moreover, regarding IGF-1, you mention in the Discussion section that ‘’most studies, including ours, have not demonstrated any correlation between IGF-1 levels and disease duration or severity’’. However, you did not correlate the IGF-1 values with any clinical parameter (disease duration or severity) and in Table 2 the IGF-1 values in the PD group were significantly higher compared to controls. The same observations are valid for IL-6, where there was also no correlation between serum values and clinical parameters performed.
Response 2: Thank you for this valuable feedback. In fact, we had performed correlation analyses between IGF-1 and IL-6 levels and clinical parameters such as disease duration, UPDRS, and MHYS scores; however, since none of these correlations reached statistical significance, we had not included them in the initial version of the manuscript. In accordance with your suggestion, we have now added the results of these correlation analyses to the revised table. Although IL-6 and IGF-1 levels appeared elevated in the PD group compared to controls, correlation analyses showed no significant associations with clinical parameters. This has now been clearly stated both in the table and within the Results and Discussion sections.
Reviewer Comment 3: The Discussion section makes multiple referrals to the lack of statistical difference of several inflammatory parameters between PD and control group. However, the statement is always extended to (see Italics): NHR was significantly higher in the PD group, whereas MHR showed no significant difference between groups. Furthermore, the lack of association between NHR and MHR with disease severity suggests that these markers provide limited predictive insight into disease progression. In our study, neither SII nor SIRI levels differed significantly between PD patients and HC, nor were they associated with disease duration, clinical stage, or symptom severity. From the presented tables and figures, I could agree that several parameters do not differ between groups, however you should not make any assumptions regarding correlations with disease duration/clinical stage, since this was not the object of your comparisons.
Response 3: Thank you very much for this insightful comment. We agree that the original wording could be misleading, as it included reference to disease duration, which was not specifically analyzed in correlation with inflammatory parameters. However, as UPDRS and HY (MHYS) scores were included in our correlation analyses and are widely accepted measures of disease severity and clinical stage in Parkinson’s disease, we have retained the statements related to these parameters. The sentence referring to disease duration has been removed to avoid overinterpretation. The results of the correlation analyses with UPDRS and HY scores are now clearly presented in the newly added correlation table. We appreciate your constructive feedback, which helped us improve the clarity and accuracy of our manuscript.
Reviewer Comment 4: Since you removed Figure 2, then the following statement should have also been removed from the Methods section - "Receiver Operating Characteristic (ROC) curve analysis was used to assess the diagnostic performance of the relevant inflammatory markers.
Response 4: Thank you for pointing this out. The statement regarding ROC analysis has now been removed from the Methods section, as Figure 2 was excluded during the revision. We have double-checked the manuscript to ensure consistency between the figures and corresponding text.
Reviewer Comment 5: Formatting: The journal template should be used for submissions.
Response 5: We appreciate your reminder regarding formatting. The manuscript has now been carefully revised to fully comply with the journal’s formatting guidelines and template requirements.
Round 3
Reviewer 3 Report
Comments and Suggestions for Authors
Dear Authors,
Although you tried to better explain the results, my concerns regarding the presentation and interpretation of data remain valid. I consider the statistical analysis too simplistic for the assumptions made in the Discussion and Conclusion section.
Author Response
Comments : Although you tried to better explain the results, my concerns regarding the presentation and interpretation of data remain valid. I consider the statistical analysis too simplistic for the assumptions made in the Discussion and Conclusion section.
Response:
Thank you for your valuable feedback and for pointing out the need for a more comprehensive statistical analysis. In response to your concern regarding the simplicity of our previous statistical approach, we have revised and expanded our analysis in the revised manuscript.
While no significant associations were initially found between UPDRS total scores and inflammatory markers, we conducted additional subgroup analyses based on UPDRS subcomponents. These analyses revealed significant correlations between inflammatory parameters and both motor and non-motor symptom domains. These findings are now presented in the newly added table (highlighted in yellow) and discussed accordingly in the revised manuscript.
We believe this refined analysis provides a deeper and clinically meaningful understanding of how specific inflammatory markers may relate to distinct clinical dimensions of Parkinson’s disease. We appreciate your guidance, which helped us enhance the robustness and interpretability of our findings.